# Coxsackievirus B3—Its Potential as an Oncolytic Virus

**DOI:** 10.3390/v13050718

**Published:** 2021-04-21

**Authors:** Anja Geisler, Ahmet Hazini, Lisanne Heimann, Jens Kurreck, Henry Fechner

**Affiliations:** 1Department of Applied Biochemistry, Institute of Biotechnology, Technische Universität Berlin, 13355 Berlin, Germany; a.geisler@tu-berlin.de (A.G.); l.heimann@campus.tu-berlin.de (L.H.); jens.kurreck@tu-berlin.de (J.K.); 2Department of Oncology, University of Oxford, Oxford OX3 7DQ, UK; ahmet.hazini@oncology.ox.ac.uk

**Keywords:** Coxsackievirus B3, microRNA, miR, oncolytic virus, cancer, virus adaptation

## Abstract

Oncolytic virotherapy represents one of the most advanced strategies to treat otherwise untreatable types of cancer. Despite encouraging developments in recent years, the limited fraction of patients responding to therapy has demonstrated the need to search for new suitable viruses. Coxsackievirus B3 (CVB3) is a promising novel candidate with particularly valuable features. Its entry receptor, the coxsackievirus and adenovirus receptor (CAR), and heparan sulfate, which is used for cellular entry by some CVB3 variants, are highly expressed on various cancer types. Consequently, CVB3 has broad anti-tumor activity, as shown in various xenograft and syngeneic mouse tumor models. In addition to direct tumor cell killing the virus induces a strong immune response against the tumor, which contributes to a substantial increase in the efficiency of the treatment. The toxicity of oncolytic CVB3 in healthy tissues is variable and depends on the virus strain. It can be abrogated by genetic engineering the virus with target sites of microRNAs. In this review, we present an overview of the current status of the development of CVB3 as an oncolytic virus and outline which steps still need to be accomplished to develop CVB3 as a therapeutic agent for clinical use in cancer treatment.

## 1. Introduction

Oncolytic viruses (OV) are replication-competent viruses which represent a promising new class of anti-cancer agents that selectively replicate in cancer cells without harming normal cells or tissues [1]. Although OV have the ability to infect both normal and cancerous cells, certain functional abnormalities of the cancer cells promote viral replication. One of these abnormalities is an enrichment of viral receptor molecules on the surface of cancer cells, which improves viral entry [2,3,4,5], while others involve defects in the cellular antiviral defense mechanisms induced by the malfunction of type I interferon pathway signaling [6,7,8], the Janus kinase (JAK)/STAT signaling pathway [9] and Protein Kinase R (PKR) activity [10,11].

Two key and closely linked mechanisms are responsible for the ability of OV to combat cancer. First, OV kill cancer cells directly as result of lytic viral replication. Viral replication within the tumors cells and tumor cell lysis leads to the release of cytokines, pathogen-associated molecular patterns (PAMPs), danger-associated molecular patterns (DAMPs) and tumor-associated antigens (TAAs), including neo-antigens [12]. These factors trigger, as a secondary anti-cancer mechanism, the induction of a systemic antitumor immunity, which includes innate and adaptive immune responses. Importantly, the immune response is also directed against distant non-infected cancer cells, thus explaining the potency of OV for the treatment of metastatic cancer disease [13,14,15].

Meanwhile a dozen OV have been evaluated or are under consideration in clinical trials against a large number of different cancers [16]. Two of them, Onocrine [17], an E1B‑deleted adenovirus, and Talimogene laherparepvec (T-Vec) [15], an attenuated herpes simplex virus 1 which has been genetically engineered to express granulocyte macrophage colony stimulating factor, have already been approved for clinical use. Clinical studies confirmed the safety of OV in cancer patients, with only low-grade adverse events observed in most cases. Regarding their efficacy, a recent published study evaluated data from more than 3200 patients treated with different OV in clinical trials between 2000 and 2020. An overall response rate (complete or partial response) of about 9% was found, and another 12% of the patients had stable disease [18]. Despite these encouraging data, it is clear that currently only a minority of patients profit from treatment with OV. Thus, the further development of OV is necessary, which also includes exploring the oncolytic activity of viral species which have not yet been investigated for their oncolytic potential.

The oncolytic activity of CVB3 was first described in 1957 [19], but similar to many other viruses whose oncolytic activity was discovered at that time, CVB3 was not seriously investigated in the following decades to determine its potential as an anti-cancer agent. However, in 2012 the oncolytic properties of CVB3 were finally seriously considered, when Miyamoto et al. [20] described the strong anti-tumor efficiency of CVB3 in a mouse model of lung cancer. In the meantime, several studies have addressed the efficiency, safety and underlying mechanisms of anti-tumor activity of CVB3 in vitro and in preclinical studies. Furthermore, a first clinical trial was initiated to assess the safety and anti-tumor potential of oncolytic CVB3 in cancer patients [21].

## 2. CVB3 Structure, Genome and Protein Functions

CVB3 is a non-enveloped single-stranded RNA virus belonging to the genus *Enterovirus* of the picornavirus family. As with all members of the *picornaviridae*, CVB3 is characterized by an icosahedral capsid of approximately 30 nm diameter, which houses the positive-sense (+) RNA genome [22,23]. The capsid consists of twelve pentamers, each composed of five asymmetric units of the structural proteins VP1–VP4 (Figure 1A). VP1 to VP3 form the viral shell. VP4 lies at the inner surface of the viral shell making a connection between *N*-termini of the other capsid proteins and the viral RNA, thereby acting as a stabilizer of the capsid pentamers during virus assembly [24,25].

The capsid surface forms a depression, called the canyon, around the five-fold axis of symmetry of each pentamer [26] (Figure 1B). Underneath the bottom of the canyon there is a hydrophobic pocket hosting a C_16_ fatty acid which is referred to as the pocket factor and contributes to the stability of the viral capsid [27,28,29]. It is thought that the binding of the Coxsackievirus and Adenovirus Receptor (CAR) [30,31] to the pocket displaces the pocket factor, thereby destabilizing the capsid, triggering the uncoating and delivery of the viral RNA into the cells [28,29,32]. Another important structural feature of the capsid surface, the elevated hypervariable puff region, located at the southern rim of the canyon (Figure 1B), functions as a known antigenic site [26,33,34]. Furthermore, it is involved in the binding of the decay accelerating factor (DAF) which serves as a co-receptor of CVB3 [35,36].

The positive-sense (+) RNA genome of CVB3 has a length of approximately 7.5 kilobase pairs (kb). It comprises a single large open reading frame (ORF) flanked by a 742 nucleotide (nt) long 5′-untranslated region (5′-UTR) and an about 100 nt long polyadenylated 3‘-UTR [22]. Particularly the long 5′-UTR builds a number of stem-loop structures, among them the cloverleaf (CL) and the internal ribosomal entry site (IRES) which play major roles in viral replication and protein synthesis [22,37,38,39,40]. The CL interacts with VPg (virus protein genome-linked, also known as 3B), which is covalently attached to the 5′-end of the positive-sense RNA, and with the 3′-UTR and other trans-acting proteins to form the replication complex during RNA synthesis [39,41,42,43,44,45]. The IRES conveys the CAP-independent interaction with the cellular ribosome for viral translation [38,41]. The ORF encodes a continuous polyprotein which is autocatalytically processed into the 4 structural (VP–VP4) and 7 non-structural proteins (2A–2C, 3A–3D), as well as 3 intermediate cleavage products (2BC, 3AB and 3CD) [41] (Figure 2).

The non-structural proteins function to promote viral protein synthesis, replication, release and spread by interacting with the RNA genome and polyproteins, while also interfering with cellular processes. Most of the manipulation of host cell processes and virus-induced pathogenesis can be traced to the activities of viral proteases 2A and 3C. Besides the proteolytic processing of the polyprotein into the 11 structural and non-structural proteins, the proteases are involved in the shutdown of host cell translation and transcription, disruption of the cytoskeleton, induction of apoptosis and attenuation of the innate immune response. The blockage of translation is mainly carried out by cleavage of host factors like the eukaryotic initiation factor 4G [46], the poly(A)-binding protein [47] and the Death-Associated Protein 5, as they are important mediators of cap-dependent and IRES-dependent translation initiation in the cell [22,48,49]. To prevent premature viral clearance from the cell, the proteases also cleave the immune adaptor molecules and pro-apoptotic factors, named Mitochondrial Antiviral Signaling Protein (MAVS) and Toll/IL-1 Receptor Domain-containing Adaptor Inducing Interferon-β (TRIF), which leads to an attenuated type I interferon response and apoptotic signaling during the early stages of CVB3 infection [50]. In addition, Protease 2A cleaves the cytoskeletal protein Dystrophin, an event shown to be important for the pathogenesis of CV-induced cardiomyopathies [51]. Another key feature of the 2A and 3C proteases is their ability to induce apoptosis through caspase-8-mediated activation of caspase-3 and to activate the intrinsic mitochondria-mediated apoptosis pathway during the late phase of viral infection [46].

The Viroporin 2B and its precursor 2BC build homo- and heteromultimers, which integrate into the membranes of the Golgi apparatus and the endoplasmic reticulum (ER) [52,53,54]. The resulting pore formation leads to a leakage of Ca^2+^ into the cytoplasm [52,53,55], disturbing pro-apoptotic signaling during the early stages of infection [53,56,57] thereby preventing a rapid clearance of the virus. Furthermore, the membrane interaction of 2B is thought to induce the formation of vesicles which are important for viral replication and release [55,56,58]. In addition, the 2C protein possesses a RNA helicase function in enteroviruses [59] which could also be confirmed for CVB3 [60,61].

The proteins 3A, 3AB and 3D interact with the viral genome to form the replication complex [42]. The 3B protein serves as the primer for the transcription of the viral genome [62]. The binding of 3AB is thought to activate the protease activity of CL-bound 3CD precursor to release the 3D polymerase and mediate the cyclization of the genome by interacting with the 3′-UTR during (−) RNA synthesis [43,63,64].

## 3. CVB3 Infections in Humans and in Experimentally Infected Mice

CVB3 usually induces mild self-limiting disease with flu-like symptoms in humans. Under certain circumstances, which involve genetic and individual predispositions, severe disease can result. Most frequently, patients suffer from aseptic meningitis, encephalitis, and myocarditis [65,66], whereas pancreatitis [67,68] and hepatitis [69] are less frequently observed. Individuals of all ages and either sex can be infected with CVB3 [70], but infants are particularly at risk. CVB3 infection in infants can lead to severe systemic disease and death by hepatic, cardiac or multi-organ failure [71,72,73].

In mice, the pancreas and heart are the main target organs of CVB3, but in contrast to humans, the pancreas is the most susceptible organ in mice [68,74,75]. CVB3 infections of the pancreas results in acute pancreatitis with advancing destruction of the exocrine part of the organ. Myocardial infection leads to direct acute and chronic inflammation, impaired cardiac contractility and heart failure [68,76]. The degree of infection, inflammatory processes and tissue damage, however, depends on several factors, such as the virulence of the virus strain, genetic background of the mice, age, sex and route of virus administration [68,74,77,78,79,80].

## 4. CVB3 Receptors and Its Importance for CVB3 Targeting of Cancer

Occurrence of viral receptors on the cell surface is a key feature that contributes to virus tropism. Hence, the expression of CVB3 receptors on cancer cells is vital for the successful treatment of cancer with oncolytic CVB3. The main receptor for CVB3 binding and uptake is CAR [31,81,82] (Figure 3 and Figure 4), a transmembrane protein which is involved in cell adhesion and inflammation [83]. In addition to CAR, several CVB3 strains, such as RD and HA, use DAF, which is involved in the regulation of complement activation and cell signaling. DAF functions as co-receptor for CVB3 attachment to the host cell surface [84,85]. The binding of DAF alone, however, is not sufficient to mediate viral entry into the cell and subsequent lytic infection [29,85]. Thus, cancer cells that express DAF but not CAR are not vulnerable to oncolytic CVB3.

CAR is expressed in many tissues, including heart, lung, liver, testis, pancreas and kidney [87,88]. It is highly expressed during fetal development and in young individuals, while it is downregulated in adults [89]. In cancer, CAR is differentially expressed. Compared to normal tissues in lung cancer, cervical cancer, endometrial cancer, ovarian cancer and urinary bladder cancer, for example, CAR appears to be upregulated, whereas in colon and prostate cancers, as well as subtypes of renal cell cancers it is strongly downregulated [90].

Two studies, investigating lung [20] and endometrial cancer [91], found a good correlation between sensitivity of the cancer cell line to oncolytic CVB3 and their CAR and DAF expression levels. In another study, however, there was no clear correlation between abundance of CAR and susceptibility of colorectal carcinoma cell lines to oncolytic CVB3 [92], which may mean that under certain conditions post-entry mechanisms may be of particular importance for cytolytic activity of oncolytic CVB3.

In addition to CAR, it has been shown that CVB3 can use heparan sulfates to enter cancer cells. Thus far, however, this has only been shown for the CVB3 variant PD, which uses *N*- and 6-*O*-sulfated heparan sulfates to infect cells [92,93,94] (Figure 3 and Figure 4). Heparan sulfates are linear polysaccharides, which consist of repeating disaccharides bound to a core protein which links them to the cell surface. Based on the analysis of the expression of the heparan sulfate D-glucosaminyl 6-*O*-sulfotransferase-2 (HS6ST2), which catalyzes the transfer of sulfate groups to the C-6 (exocyclic carbon) of the glucosamine residue in heparan sulfate proteoglycans, the stomach, liver, adrenal gland, bronchus, breast, ovary, uterus, kidney and skin contain *N*- and 6-*O*-sulfated heparan sulfates. In other organs, such as the lung, pancreas, heart, spleen, prostate and colon, HS6STS expression could not be detected [95]. HS6ST2 is also differentially expressed in cancer. This enzyme is downregulated in ovarian cancer [96] but overexpressed in colorectal, gastric and pancreatic cancer [95,97,98].

A recent study from our group confirmed the importance of *N*- and 6-*O*-sulfated heparan sulfates for infection of cancer cells with the CVB3 variant PD. In fact, there was a positive correlation between expression HS6ST2 and the sensitivity of colorectal cancer cell lines to the PD strain of CVB3 [92].

## 5. CVB3 Strains, Their Oncolytic Activity and Treatment-Related Side Effects

Since CVB3 has the potential to induce severe disease, it is important to carefully select CVB3 strains for the treatment of cancer, which can infect tumor cells but have no or only low affinity and toxicity in normal tissues. The prototype of CVB3 was isolated about 70 years ago from a three-year-old child, named Nancy [99] and much of our knowledge about the mechanisms and progression of CVB3 disease in humans comes from animal studies in mice using “Nancy” and Nancy-related strains. In the last several decades, various CVB3 strains have been isolated from patients and some of them were subsequently adapted in vitro or in vivo to specific environments [68,91,100,101,102].

Based on their virulence in mice, CVB3 strains can be divided into three groups: avirulent which means with no pathologic changes in either pancreas or heart; pancreovirulent, which induces pathologic changes only in the pancreas but not in heart; cardiovirulent, with pathologic changes in both the pancreas and heart [68]. Strains of CVB3 belonging to each of these groups have been tested for their oncolytic activity. The most commonly used strain is the Nancy strain (Figure 5, Table 1). As found in several studies, this strain has a potent cytolytic activity against a variety of cancer cell lines in vitro, including human non-small cell lung cancer and small-cell lung cancer, cervical carcinoma, pancreatic carcinoma, colon carcinoma and breast carcinoma [20,91,92,103,104,105]. In vivo, the Nancy strain showed an impressive oncolytic activity in xenografted mouse models of non-small cell lung cancer [20,103], small cell lung cancer [105], Kirsten rat sarcoma viral oncogene homolog (*KRAS*) mutant lung adenocarcinomas [104], colorectal carcinomas [92] and in syngeneic mouse models of lung cancer [20,103]. However, these studies revealed virus-induced toxicity. Pathological alterations in mice particularly involving the heart and pancreas and were characterized by the destruction of the exocrine part of the pancreas and myocardial injury and inflammation. The severity of the virus-induced disease varied among the studies. In some studies, animals had to be sacrificed for severe morbidity [92,104,105], whereas other studies observed no treatment-related deaths and only moderate pancreatitis, hepatic dysfunction and mild myocarditis [20,103]. The reasons for different toxicity are not obvious but can possibly be traced to the use of variants of the Nancy strain, which differ in virulence or/and the specific modalities of experimental procedure.

CV-B3/2035A is another CVB3 strain which was recently evaluated for its oncolytic activity (Figure 5, Table 1). This strain was isolated from a patient with hand, foot and mouth disease and differs clearly from the Nancy strain. Its nucleotide sequence is only about 80% identical to the Nancy strain, though its amino acid identity is about 95% [91]. The virus exerts significant therapeutic effects against subcutaneous human endometrial tumors after either intratumoral or intravenous administration in nude mice. Moreover, it caused a 10–40% loss in viability of patient-derived endometrial cancer tissue biopsies ex vivo, showing that cancer tissue from patients is susceptible to CV-B3/2035A. Regarding toxicity in mice, there was no mortality or pathological changes in the brain, heart, liver, spleen, lung and kidney, but virus RNA was detected in the heart. The pancreas, however, was not investigated, so that a final assessment of the toxicity of the virus in vivo is still pending.

Our group applied the CVB3 variant PD to the treatment of colorectal carcinomas. PD is a derivative of the laboratory CVB3 variant *p* [102] (Figure 5) and induces lytic infection in primary human fibroblasts (HuFi H), which is not seen for other CVB3 strains [106]. The virus has a unique receptor tropism. Due to several mutations within the VP1 gene (Figure 3), PD can interact with *N*- and 6-*O*-sulfated heparan sulfates, thus enabling its entry into CAR-deficient target cells [94,106]. However, it also recognizes DAF and can also use CAR as an entry receptor. PD was more efficient at killing colorectal cancer cells in vitro than other CVB3 strains, such as Nancy, H3, and 31-1-93. After intratumoral injection, the virus markedly reduced colorectal cancer growth in a xenografted mouse model. Importantly, this occurred without affecting normal organs, including the pancreas and heart [92]. The reasons for the complete attenuation of the virus in normal organs observed in our and in other studies [77,92] is not clear, but lack of sufficient amount of *N*- and 6-*O*-sulfated heparan sulfates on cell surface of normal tissues [95] and low affinity of the virus to CAR [106] may play a role. On the other hand, rare cases of pancreatic and cardiac toxicity in individual PD-treated animals have been reported [92]. This, however, was related to the emergence of a mutant of PD after in vivo application of PD.

The CVB3 strains H3 and 31-1-93 are highly toxic to mice (Figure 5, Table 1). H3 is a cardiac-adapted laboratory strain of Nancy [100] and 31-1-93 is a derivative of PD obtained after four passages in the mouse heart [102]. Although they possess potent oncolytic activity, each kills nude mice a few days after intratumoral virus injection [92,107] and induces severe pancreatitis and myocarditis in immunocompetent mice [74,77].

## 6. Influence of Oncolytic CVB3 on the Tumor Microenvironment

One of the most important features of OV is that they induce immunogenic cell death after they infect the cancer cells. Shedding of DAMPs and PAMPs into the tumor microenvironment (TME) stimulates the immune system to act against the viruses and, more importantly, against the cancer cells. This also induces activation of systemic immunity against disseminated tumors and provides protection from tumor relapse. Additionally, OVs turn immunosuppressed “cold” tumors into the immunogenic “hot” tumors by infecting immunosuppressive cells or converting them into pro-inflammatory phenotypes [111]. Due to these important abilities, OVs have the potential to be combined with other immunotherapies, such as checkpoint inhibitors or adoptive T cell therapy, which may lead to synergistic effects under the right circumstances [112].

The most commonly investigated parameters to determine immunogenic cell death are extracellular ATP, high-mobility group box 1 (HMGB1) and cell surface calreticulin levels [113]. Miyamoto et al. documented that the infection of NSCLC cells with CVB3 led to the high expression of cell surface calreticulin, elevated levels of extracellular ATP and HMGB1 translocation in vitro. Moreover, in vivo analysis, using an athymic mouse model, revealed that CVB3 infected tumors had elevated levels of dendritic cells (DCs), granulocytes and NK cells. They concluded that cytotoxic effect of NK cells had a crucial role in tumor regression, as the depletion of these NK cells abolished the therapeutic effect. While this study provides an important insight into understanding CVB3-induced antitumor immunity, it does not show the complete picture due to lack of adaptive immunity in this animal model. Adaptive antitumor immune response activation in immunocompetent mouse models has not been shown for any of the tested oncolytic CVB3 strains.

Another indicator of an activated systemic antitumor immune response is the abscopal effect. This phenomenon is related to the detection of the therapeutic effect in the untreated distant tumor lesions, along with the expected effect in the treated lesions [13,14]. As well as using metastatic animal models, the abscopal effect can be measured in mice bearing subcutaneous tumors on bilateral flanks by infecting one tumor and leaving the other one untreated. The shrinkage of untreated contralateral tumor following oncolytic CVB3 treatment has been reported in several studies [20,91,92,104]. Of note, these observations were based on athymic mouse models. While it has been suggested that, in nude mice, immune system-mediated abscopal effect can be induced by cytokines, innate immune cells or even T cells [114]; in oncolytic CVB3 studies, the regression of distant tumors was associated with viremia [20,91,92,104]. Distinctively, Hazini et al., by taking advantage of in situ hybridization, detected viral RNA in tumor-infiltrating immune cells [92], suggesting that viruses might be carried to the other tumor sites via infected immune cells. Indeed, several reports for other OV have shown that immune cells can carry viruses to distant sites while protecting them from neutralizing antibodies and the complement system [115,116,117]. Such a mechanism has not yet been shown for oncolytic CVB3, though it certainly seems plausible. As a matter of fact, the interaction between immune cells and CVB3 with respect to the potential for virotherapy has not yet been investigated.

Antigen presenting cells, particularly dendritic cells, represent a crucial immune cell component in the stimulation of adaptive immunity. They orchestrate the immune system and form a bridge between the innate and the adaptive immune responses. Therefore, it has become important to examine the interaction between OV and dendritic cells. Although no such studies have yet been performed for oncolytic CVB3 strains, Kemball et al. showed that the CVB3 strain H3 was not able to infect and cause cytopathic effects in murine or human DCs in vitro [118]. On the other hand, intriguing studies have been reported for other oncolytic picornaviruses, such as poliovirus and Coxsackievirus A21 (CVA21) [119,120]. Infection of DCs with one of the viruses was not lethal, however, it led to the secretion of cytokines and subsequent activation of tumor-antigen specific T-cells. This important feature was also linked to the potentially synergistic effect of small RNA viruses with immunotherapeutics, such as checkpoint inhibitors [121].

## 7. Improvement of the Safety of Oncolytic CVB3 by MicroRNA-Mediated Regulation of Virus Replication

Since oncolytic CVB3 can cause severe side effects, improving the safety of oncolytic CVB3 is a necessary requirement before the virus can be used for clinical applications. One of the most promising approaches is the engineering of CVB3 to become sensitive to microRNAs (miRs) expressed in tissues where virus replication must be prevented. MiRs are endogenously expressed small noncoding RNAs that post-transcriptionally regulate gene expression [122]. Many miRs are expressed differentially in tissues and organs [123,124,125]. Cancer cells also differ in their miR expression profile compared to healthy tissues [126,127]. To make OV sensitive for miRs, they are equipped with miR target sites (miR-TS) which correspond to miRs that are abundantly expressed in healthy tissues but poorly expressed or absent in tumor cells. That is to say that the virus is prevented from replicating in healthy cells due to a corresponding miR, which selectively recognizes and eliminates the viral RNA containing the miR-TS in healthy, but not in cancer cells, thus de-targeting the virus from healthy tissues [128]. Typically, two to four copies of a miR-TS are inserted into the viral genome and the sequence of the miR-TS is completely complementary to the corresponding endogenously expressed miR [128,129]. The latter ensures endonucleolytic cleavage of miR-TS-containing RNA by Argonaute 2 [130,131]. In RNA viruses equipped with miR-TS, virus replication is not only suppressed by miR-mediated downregulation of viral protein expression, but also by direct destruction of the viral genome. Therefore, this class of viruses seems to be particularly sensitive to this approach.

Several studies have demonstrated that OV with miR-TS efficiently de-targets the virus from healthy tissue without affecting their oncolytic activity in cancer [128,132,133,134,135,136]. The first successful attempt to selectively inhibit CVB3 with miRs was carried out in 2015 by He et al. [137]. This study showed that insertion of miR-TS of muscle-specific miR-206 and miR-133 into the CVB3 genome strongly reduced virus replication in the heart and thereby decreased CVB3-mediated heart pathology, which increased overall animal survival. In addition to these encouraging results, the study identified several important limitations related to the insertion sites of miR-TS into the CVB3 genome. In fact, the virus could only be propagated from an infectious cDNA clone if miR-TS were inserted into the 5′-UTR close to the codon for translation initiation. In contrast, the insertion of miR-TS into the 5′-UTR upstream (nucleotide position 249) of the core sequence of the IRES (located between nucleotide 432 and 639 within the CVB3 genome [138]) as well as insertion of miR-TS in the 3′-UTR at nucleotide positions 7387 or 7359 to 7360 were not tolerated by the virus (Figure 6). As a possible reason for these observations, He et al. suggested that the miR-TS probably disturbed the higher-order RNA structure of the viral genome [137]. This assessment is in line with another report showing delayed propagation of CVA21 from infectious viral RNA after insertion of miR-TS into a stem loop structure of the 3′-UTR which is important for viral RNA synthesis and poly(A) tail elongation [139]. Nevertheless, several studies recently demonstrated that miR-TS can also be inserted into the 3′-UTR of CVB3 without affecting virus propagation. This was achieved by placing the miR-TS immediately downstream of the stop codon of the CVB3 polyprotein [74,103,107,108,109]. This approach apparently prevents the destruction of critical secondary structures within the 3′-UTR of the virus.

It has also been shown that miR-TS can be placed into the 5′-terminus of the CVB3 polyprotein encoding sequence. However, virus propagation in target cells, as well as inhibition of virus replication and cytotoxicity in cells expressing corresponding miRs were about 10-fold lower compared to CVB3 containing the miR-TS in the 3′-UTR [109]. Similar observations were made when comparing CVB3 with miR-TS in the 5′-UTR and 3′-UTR of the viral genome. Jia et al. [103] observed lower pathology in the pancreas of mice when the CVB3 strain used contained miR-TS of the miR-34a or miR-34c in their 3′-UTR rather than in their 5′-UTR. This indicates that miR-TS within the 3′-UTR are better recognized by the corresponding miR than when they are within the 5′-UTR. This observation was explained by the easier dissociation of the RNA-induced silencing complex from 5’ miR-TS of the RNA when ribosomes bind this region during the initiation of translation [140].

The significance of the orientation of miR-TS within the CVB3 has also been investigated. This involved the insertion of miR-TS in a forward orientation to target the plus-strand genome or in a reverse orientation to target the minus-strand replication intermediate of CVB3. All CVB3 containing miR-TS in their plus-strand genome were sensitive to the corresponding miRs [74,103,105,107,108,109,137]. The miR-targeting of the viral minus-strand is particularly attractive, as CVB3 generates only few minus-strand RNA intermediates during genome replication, which function as template for generating lots of plus-strand RNA genomes [141]. However, targeting the minus-strand by miRs failed to inhibit viral replication [109] or merely showed a very low level reduction in viral replication compared to targeting the positive-strand genome [105]. The reason for the failure has been suggested to be the inaccessibility of the CVB3 minus-strand in the viral replication complexes to the RNA interference machinery [142].

The stability of miR-TS in the CVB3 genome is an essential requirement for the safety of oncolytic miR-regulated CVB3. In CVB3 this is of particular importance because of the error-prone viral RNA polymerase [143,144,145] leading to high mutation frequency within the viral genome of 10^−4^ to 10^−5^ [146]. To investigate the occurrence of mutated miR-TS in oncolytic CVB3, we sequenced the miR-TS of two oncolytic strains of CVB3 from virus-injected tumors at day 32 after virus injection. Single nucleotide substitutions or small stretches of up to three nucleotide substitutions were detected in some of the copies of the miR-TS. However, at least one copy remained completely intact, which was obviously sufficient to prevent undesirable virus replication and organ-toxicity [107]. In another study Liu et al. [105] observed complete loss of miR-TS from an oncolytic CVB3 in some animals 35 days after virus injection resulting in severe side effects in the affected animals. As mentioned by the authors, this effect was probably provoked by formation of stem loop structures as result of the cloning strategy of the miR-TS, which facilitated the removal of the miR-TS during viral replication. If so, strategies to minimize or eliminate this possibility must be developed and used.

A further important question concerns the strength of virus attenuation achieved by miR-mediated suppression of virus replication. In vitro data have shown an impressive reduction in miR-TS-bearing CVB3 replication of up to 10^5^-fold in cells endogenously expressing their cognate miRs, which was corroborated in vivo. Even after the application of high doses of virus and the use of highly virulent virus strains, a complete to nearly complete abrogation of CVB3 replication and organ toxicity in tissues expressing corresponding miRs was observed in several studies [74,103,105,107,108,109]. However, it should be mentioned that the degree of attenuation seemed to be affected by several factors, such as the absolute level of miR expression in normal tissues [107], the miR-TS copy number [103] and the route of viral application [105].

All studies published to date aimed at improving the safety of oncolytic CVB3 by miR regulation focused on abrogation of CVB3 toxicity in the pancreas and/or the heart, as these are the most CVB3-sensitive organs in mice. Hazini et al. [107] inserted miR-TS of miR-375, which is specifically expressed in the pancreas, and miR-1, which is specifically expressed in the heart, into the highly pancreato- and cardiotropic CVB3 variant H3. In a xenograft model of colorectal carcinoma, virus replication was completely abrogated within both target tissues and both organs were protected from virus-induced pathology, while the growth of colorectal carcinomas was still significantly inhibited. Similar results were obtained by Sagara et al. [108], who generated a CVB3 strain with the miR-TS of miR- 1 and of the pancreas-specific miR-217. The virus was successfully de-targeted from the heart and pancreas of nude mice, while breast cancer growth was still significantly inhibited. Other studies used another strategy and inserted miR-TS corresponding to tumor-suppressor miRs into the CVB3 genome. Liu et al. [105] and Jia et al. [103] showed that miR-143/miR-145- and miR-34a-sensitive CVB3 had a strongly reduced virulence in the pancreas and the heart, while their oncolytic activity against lung carcinoma remained unaffected. However, although the insertion of miR-TS for tumor suppressor miRs might be the easiest way to de-target CVB3 from a wide panel of normal tissues, there are some limitations of this approach. At first tumor suppressor miRs might be variably expressed, as cancer is often heterogeneous. Thus, a miR that is downregulated in one type of cancer might be present at high levels in another one. For example, we and others found abundant expression of members of the tumor suppressor miR-34 family in colorectal carcinomas [107,147], which makes miR-34-mediated CVB3 de-targeting unsuitable for virotherapy against colorectal cancers with high expression levels of miR-34. Furthermore, suppression of the CVB3 in the pancreas and heart by tumor suppressor miRs seems to be less pronounced than suppression by tissue-specifically expressed miRs [103,107], probably because of lower absolute expression levels of these miRs. Nevertheless, this disadvantage can be overcome by inserting more copies of the tumor suppressor miR’s target sites into the CVB3 genome [103].

## 8. Directed Virus Evolution as a Strategy to Increase Anti-Tumor Efficiency of Oncolytic CVB3

Although the efficacy of oncolytic CVB3 has been documented in different cancer entities, due to the great heterogeneity, certain tumors may resist treatment with oncolytic CVB3, limiting the scope and utility of oncolytic CVB3 in cancer therapy [20,91,92,109]. Directed evolution is a strategy to increase infectivity and virus spread to counteract this issue. It is based on the ability of viruses to intrinsically adapt to tumor environments by acquiring beneficial mutations. CVB3 is well suited to this adaptation process due to its high mutation rate, which enables the adaptation of the virus within short time frames [143,144].

The first successful approach to adapt CVB3 to a tumor environment was reported in 1957 when Suskind et al. [19] demonstrated an increase in the oncolytic activity of CVB3 after passaging the virus in vivo in HeLa-derived tumors in rats. This approach is not limited to in vivo passaging, as Borderia et al. [148] demonstrated that the passage of CVB3 on permissive and less permissive cancer cells increased viral fitness. The changes acquired during adaptation could be mapped to amino acid changes within the receptor binding sites of the capsid proteins, suggesting alterations in receptor binding and affinity as the most frequent basis of the mechanism of adaptation. In accordance with this, Svyatchenko et al. [149] showed an impressive increase in the oncolytic activity of a CVB6 strain, which is closely related to CVB3, after in vitro adaptation either to RD cells or MCF7 cells. Using the adapted viruses in vivo, a complete inhibition of growth of the respective RD and MCF7 tumors was observed, whereas the parental virus had almost no inhibitory effect. Importantly, the safety of the virus seemed not to be negatively affected by the adaptation process.

## 9. Genetic Engineering of CVB3 to Enhance Its Anti-Tumor Efficiency

A widely used approach to further enhance the anti-cancer efficiency of OV is to arm them with tumor-toxic transgenes. For DNA viruses with large viral genomes such as adenoviruses or herpesviruses, the insertion of transgenes is comparatively easily, as even the insertion of large foreign sequences does not prevent viral replication or packaging [150]. CVB3 has a rather small genome and tolerates foreign gene sequences within a range of about 10% of the wild-type virus genome size, as demonstrated in recombinant CVB3 by placing a GFP reporter encoding sequence into the CVB3 genome [151,152]. However, not only the size of the transgene is important for the proper function of the virus, but also the site of transgene insertion within the viral genome and the flanking sequences, which are necessary to release the transgene from the viral polyprotein. Placing the GFP-encoding sequence at the 5′-end of the open reading frame of the CVB3 polyprotein, separated by an artificial 3Cpro/3CDpro encoding cleavage site from the VP4 encoding sequence, resulted in strong GFP expression of the virus. The recombinant viruses, however, suffered from a lower rate of proliferation, instability of the transgene and reduced virulence [151,153,154]. Similar observations were made when a murine interleukin-4 encoding sequence was inserted into the junction between the viral VP1 and 2A genes, flanked by sequences encoding identical 2Apro cleavage sites. The use of non-identical sequences of the 2Apro cleavage sites, however, improved the stability of the virus and led to sustained transgene expression in vivo [152,155]. The latter may, therefore, be suitable to arm oncolytic CVB3 with tumor-toxic transgenes.

In a first approach, to improve the efficacy of CVB3 by arming the virus, Cai et al. [110] analyzed the antitumor effectiveness of an attenuated CVB3 in a xenograft lung cancer model, after the insertion of basic peptides at the 5′-end of the CVB3 polyprotein coding sequence. Treatment of mice with the viruses was associated with significantly higher pH values within lung cancer tissues and antitumor activity appeared to be improved compared to recombinant CVB3 without the basic peptides [110].

## 10. Oncolytic Coxsackievirus B3 *Versus* Coxsackievirus A21

CVA21 is a naturally occurring enterovirus that can induce mild infections of the upper respiratory tract in humans, but also myositis [156,157,158]. The virus is one of the most studied OV. It has been developed from the wild-type CVA21 Kuykendall strain, and is under development by Viralytics Limited as CAVATAK ™. Both in preclinical cancer models and in clinical studies in humans of various cancers, CVA21 showed significant anti-cancer efficiency, while it was mostly well tolerated [159,160]. As a member of the same virus family, CVA21 has great similarities with CVB3 in terms of its morphology, replication and induction of immunological response. An important difference is the receptor tropism. CVA21 binding and uptake occurs through the intercellular adhesion molecule-1 (ICAM-1), an immunoglobulin-like molecule [161,162,163]. In addition, similar to CVB3, DAF acts as a co-receptor for binding the virus to the cell surface. However, binding to DAF does not lead to a lytic infection and is not essential for the infection of the target cells with CVA21 [163]. ICAM-1 is commonly expressed at high levels on tumor cells, which seems to be responsible for the tropism of CVA21 to a wide variety of cancers [159].

It is currently not possible to assess whether the treatment of cancer with CVB3 could be advantageous over treatment with CVA21, as comprehensive comparative experimental investigation is yet to be conducted. However, the expression of CAR and ICAM-1 may be different in tumors, so that tumors of any given patient might be more sensitive to CVB3 or CVA21. This is supported by the study of Shafren et al. [164] which showed that RD rhabdomyosarcoma cells, which express low levels of DAF and CAR, but do not express ICAM-1, were susceptible to infection by CVB3 but not by CVA21, while melanoma cells were more sensitive to CVA21 than to CVB3, because they lack CAR. In a similar approach, Miyamoto et al. [19] found different susceptibility of lung, renal and colon carcinoma cell lines for CVB3 and CVA21.

By determining individual CAR and ICAM-1 expression patterns on tumor biopsies of patients, a pre-selection for one of the viruses might be possible. However, post-entry factors also determine the susceptibility of the tumors for both viruses [92,165]. Therefore, additional screening of oncolytic activity of the viruses, for example in primary cultures of ex vivo tumors [91] or in patient-derived xenograft (PDX) tumor mouse models [166], may be prove necessary in order to be able to make an optimal selection.

## 11. Conclusions/Outlook

Research over the last few years has recognized CVB3′s potential as an OV with an impressive anti-tumor efficiency in lung, colorectal, breast and endometrial cancer in pre-clinical settings (Table 1). In addition, the broad expression of CVB3 receptors on tumor cells and the susceptibility of cancer cells from many different types of cancer to infection with CVB3 will also make the virus an attractive option for the treatment of other cancers. The anti-cancer efficacy of the virus is clearly related to key mechanisms involved in the killing of cancer cells by OV, direct virus-induced tumor cell lysis, and the induction of a strong anti-tumor immune response. In particular, the latter underscores the potential of CVB3 for utilization in the treatment of advanced metastatic cancers. The majority of oncolytic CVB3 variants tested so far are sufficiently tolerated in vivo, but side effects can still occur. To avoid such side effects, the virus can be made susceptible to tissue-specific or tumor-suppressor miRs by genetic engineering, which has proven effective at reducing or eliminating the toxicity of even highly toxic oncolytic CVB3 variants.

There are still various aspects that must be addressed in order to further develop CVB3 as an OV. Currently, oncolytic CVB3 has only been used against certain types of cancer in vivo. Expanding the investigations in the future to include other cancers could provide a broader view of the true cancer-fighting potential of the virus. Accordingly, the potential of the CVB3 virus variants that have already been used, or may be used in the future, needs to be investigated further and compared to each other to find the advantages and disadvantages of their use in cancer therapy. An increase in efficacy may be a further issue which will need to be addressed in the development of oncolytic CVB3. The anti-cancer efficacy of CVB3 can be improved by directed evolution to attack less susceptible cancers, arming the virus with tumor-toxic transgenes or combining oncolytic CVB3 with other cancer therapies such as chemotherapeutic agents and cancer immunotherapies. In addition, a better understanding of how the virus interacts with the immune system contributes to modulating TME and inducing systemic anti-cancer activity is needed. Developments in these fields will be essential to further develop oncolytic CVB3 as an oncolytic agent, which may provide a ray of hope in combating some of the deadliest cancers currently known, such as advanced metastatic cancers.

## Figures and Tables

**Figure 1 viruses-13-00718-f001:**
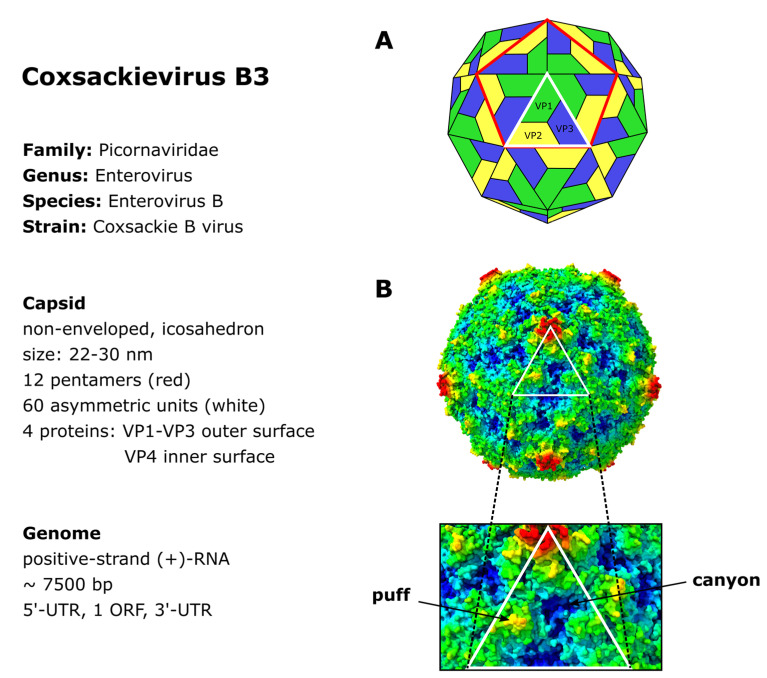
Structure of the CVB3 capsid. (**A**) A schematic model of the icosahedral CVB3 capsid structure composed of 60 asymmetric units, each composed of one VP1 (green), VP2 (yellow), VP3 (blue) and VP4. VP1, VP2 and VP3 form the capsid surface, while VP4 is located on the inside surface. The red outlined area depicts a pentamer, the white outlined triangle depicts an asymmetric unit. (**B**) *Upper panel*: The capsid surface structure colored radially. The radial surface structure was calculated and modelled with the bioinformatic software UCSF ChimeraX [23] based on the structural data of CVB3 RD from the RCSB protein databank (accession no. 4GB3). *Lower panel:* A magnified section of the asymmetric unit with the arrows highlighting the puff region and the canyon.

**Figure 2 viruses-13-00718-f002:**
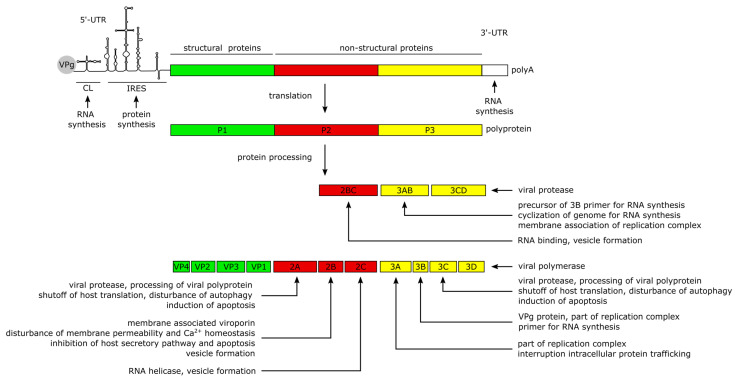
Structure of the CVB3 genome and the functions of the viral proteins. The ~7.5 kb viral genome comprises a 5′-UTR with the cloverleaf (CL) structure and the internal ribosomal entry site (IRES), modelled according to Bailey et al. [40], an open reading frame and a 3′-UTR with a poly(A) tail. The small VPg protein binds to the 5′-end of the genome. The open reading frame is translated into a polyprotein which is autocatalytically processed into the structural (green) and non-structural (red and yellow) proteins. During the processing of the non-structural proteins, three precursor proteins (2BC, 3AB, 3CD) with distinct functions in the viral life cycle are formed, which are processed further into the seven non-structural proteins (2A, 2B, 2C, 3A, 3B, 3C, 3D).

**Figure 3 viruses-13-00718-f003:**
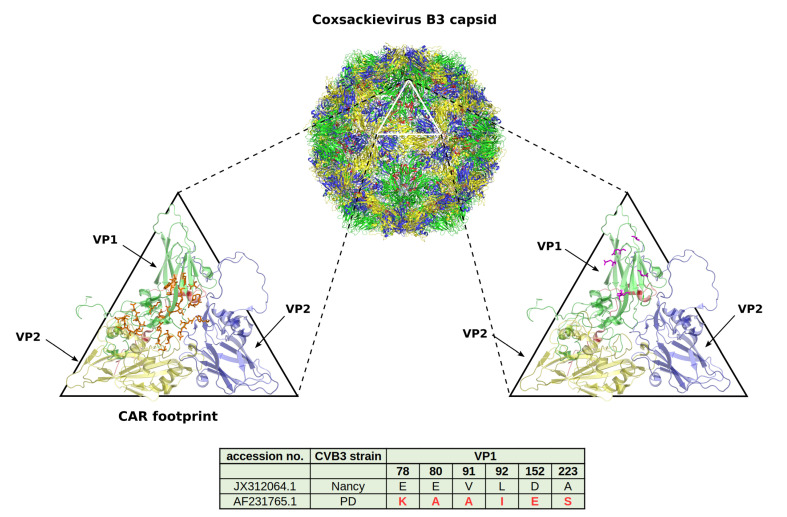
Amino acids of the CVB3 capsid involved in CAR and heparan sulfate binding. *Upper image*: The capsid is shown along with the tertiary structure of the capsid proteins VP1 (green), VP2 (yellow), VP3 (blue) and VP4 (red). The triangle outlines one asymmetric unit. The structure is modelled with the bioinfomatic software PyMOL (The PyMOL Molecular Graphics System, Version 2.0 Schrödinger, LLC) based on the structural data of CVB3 RD from RCSB protein databank (accession no. 4GB3). *Lower left magnified image*: the amino acids involved in CAR binding (according to He et al. [86] and Organtini et al. [26]) are shown in orange. *Lower right magnified image*: the amino acids within one asymmetric unit of CVB3 PD strain (NCBI accession no. AF231765.1) that differ to the CVB3 Nancy strain (NCBI accession no. JX312064.1) are shown in magenta. Their position within VP1 is indicated in the table. The amino acids which differ in the sequence of PD compared to Nancy are involved in binding of the virus to *N*- and 6-*O*-sulfated heparan sulfates.

**Figure 4 viruses-13-00718-f004:**
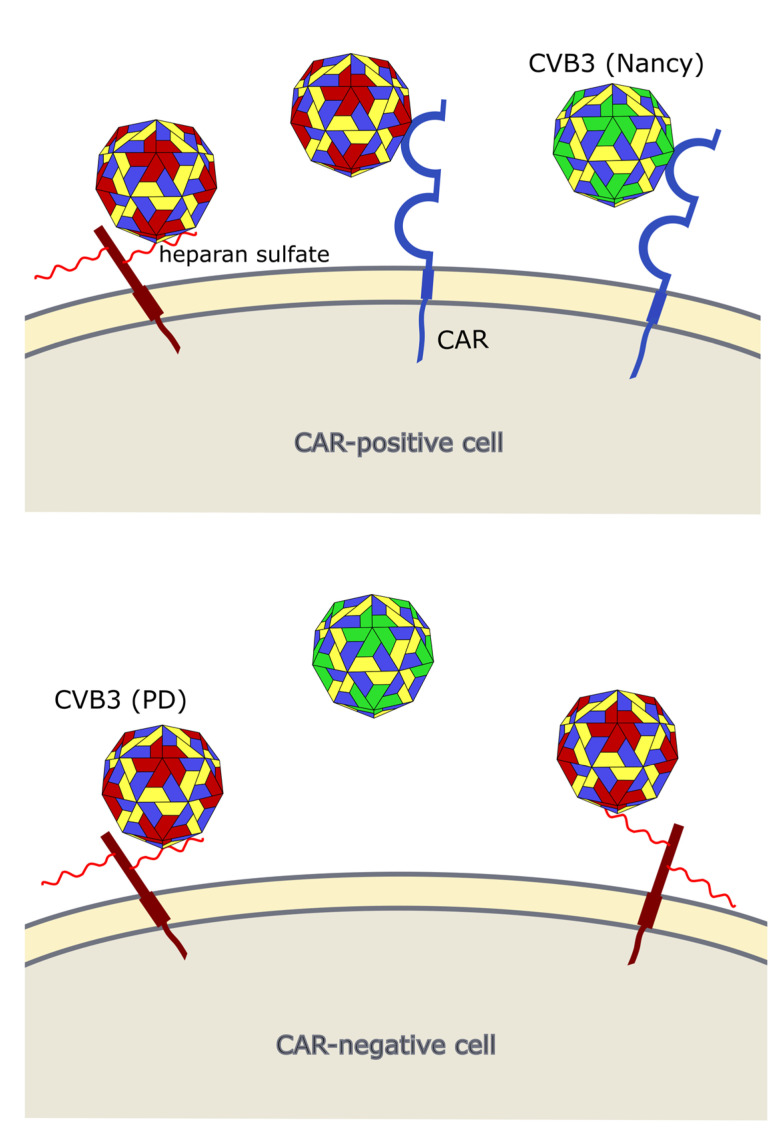
Interaction of CVB3 with cellular receptors. The prototype CVB3 strain Nancy uses CAR for cell entry, whereas the CVB3 variant PD can infect cells via *N*- and 6-*O*-sulfated heparan sulfates and CAR. *Upper panel*: CAR-positive cells can be infected with PD and the Nancy strain. *Lower panel*: CAR-negative cells cannot be infected with the Nancy strain, but with PD when *N*- and 6-*O*-sulfated heparan sulfates are expressed on the cell surface.

**Figure 5 viruses-13-00718-f005:**
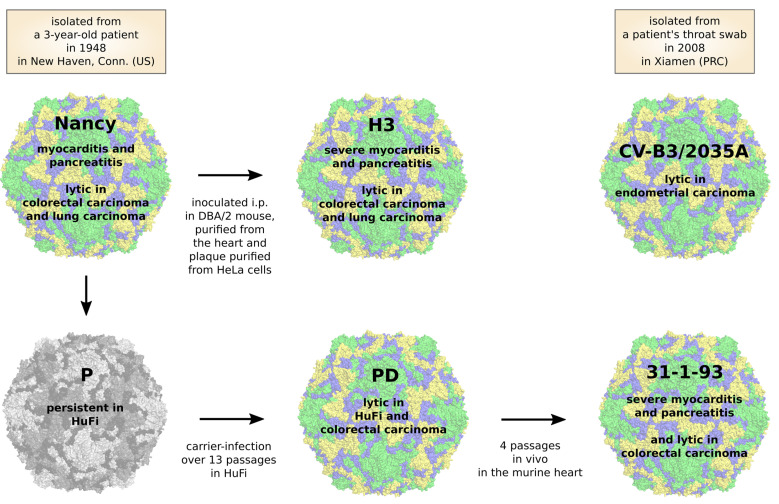
Oncolytic CVB3 strains-origin, oncolytic efficacy and side effects. The Nancy strain of CVB3 was isolated in New Haven, CT, USA in 1948 from a 3-year-old patient called Nancy [99]. The Nancy strain induces pancreatitis and myocarditis and has been shown to be lytic in colorectal and lung cancer [20,92,103,104,105]. The PD variant arose after 13 passages of persistent CVB3 Nancy *p* strain in human fibroblasts (HuFi) [102]. The PD strain is lytic in HuFi and colorectal carcinoma [92,102]. The strain 31-1-93 was isolated after passaging the PD strain through the murine heart [106]. It shows oncolytic activity in colorectal carcinoma and induces severe pancreatitis and myocarditis in mice [77,92]. CV-B3/2035A strain was isolated in Xiamen, PRC in 2008 from a patient’s throat swab, who presented with light symptoms of hand, foot and mouse disease. The CV-B3/2035A strain has oncolytic potential in endometrial carcinoma [91]. The strain H3 is a cardiotropic strain of CVB3 Nancy, which has oncolytic activity in colorectal cancer but induces severe pancreatitis and myocarditis [74,100,107].

**Figure 6 viruses-13-00718-f006:**
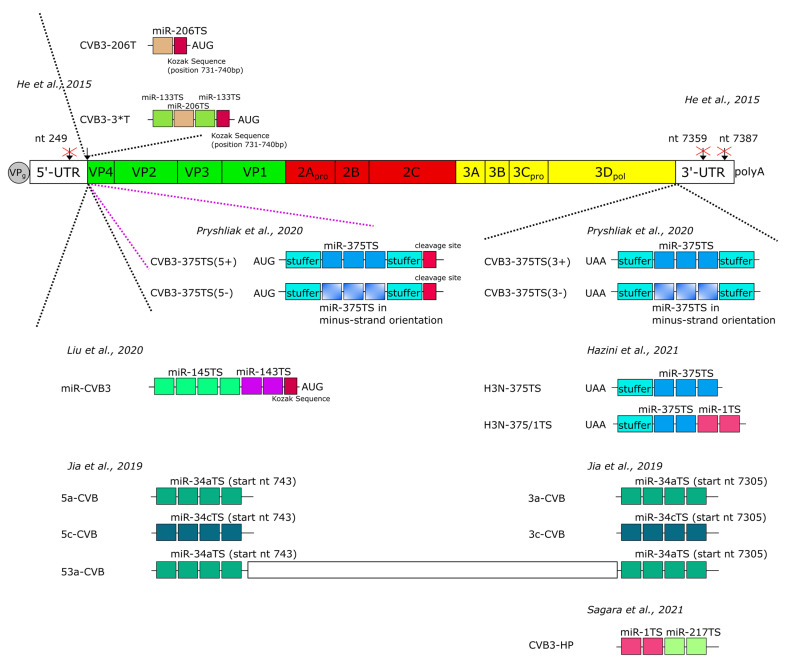
MiR-regulated oncolytic CVB3. A schematic representation of the CVB3 viral genome depicting the insertion sites and copy number of miR-TS used in publications to date. CVB3 genome sites which did not tolerate insertion of miR-TS are shown by black arrows with red crosses through them.

**Table 1 viruses-13-00718-t001:** Oncolytic CVB3 virotherapy for cancer treatment.

Target Cells for CVB3	Cancer Model; Route of Application	CVB3 Variant	MiR-Regulation		Aimed de-Targeting of	Reference
human breast cancer cells (TNBC cells)	mouse MDA-MB-468 xenograft; intratumoral	not stated	tissue‑specific	miR-1 miR-217	pancreas, heart	[108]
human colorectal carcinoma cells	mouse DLD-1 xenograft; intratumoral	PD		miR-375 miR-1	pancreas, heart	[107]
human colorectal cancer cells	colorectal cancer cells	attenuated H3		miR-375	pancreatic cells	[109]
human lung cancer cells (KRAS^mut^ lung adenocarcinoma non‑SCLC, TP53^mut^/RB1^mut^ SCLC cells)	mouse H526-derived TP53^mut^/RB1^mut^ SCLC xenograft; intraperitoneal	Nancy	tumor suppressor	miR-145 miR-143	heart, lung (lung epithelial cells, cardiomyocytes)	[105]
lung cancer cells (non‑SCLC H1299, TC-1)	mouse H1299 xenograft and TC‑1 syngeneic lung cancer model; intratumoral	Nancy		miR-34a miR-34c	normal cells	[103]
human lung cancer cells (non‑SCLC)	mouse GLC-82, A549, H460 xenograft; intravenous	attenuated Nancy—modified with basic peptide	-	-	-	[110]
human lung cancer cells (KRAS^mut^ lung adenocarcinoma non‑SCLC)	mouse H2030 xenograft; intratumoral	Nancy	-	-	-	[104]
various cancer cell lines, esp. human non‑SCLC lung cancer cells	mouse A549, H1299, EBC-1 xenograft and TC‑1 syngeneic lung cancer model; intratumoral	Nancy	-	-	-	[20]
human EC cells	mouse HEC-1-A, HEC-1-B, Ishikawa xenograft; intratumoral, intravenous	CV-B3/2035A	-	-	-	[91]
human colorectal cancer cells	mouse DLD-1 xenograft; intratumoral	PD, Nancy, H3 and 31-1-93	-	-	-	[92]

TNBC, triple-negative breast cancer; KRAS, Kirsten rat sarcoma viral oncogene; SCLC, small-cell lung cancer; RB, retinoblastoma protein; TP, tumor protein; EC, endometrial cancer.

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
