# Peer review of "Coxsackievirus B3—Its Potential as an Oncolytic Virus"

_viruses, 2021, doi:10.3390/v13050718_

Round 1

Reviewer 1 Report

Oncolytic viruses (OV) are a promising tool in the fight against cancer with two constructs already used for treatment. Numerous other OVs are in various stages of development with a constant flow of new studies adding important information on their mode of actions and describing new viral variants with improved oncolytic potency.

The manuscript by Geisler and colleagues offers a detailed view of the development Coxsackievirus B3 (CVB3) as an oncolytic agent. The article is carefully constructed, well written, richly referenced (149 references), and addresses all the aspects relevant to the use of CVB3 as an OV.

It describes the modes of actions of OVs (1), offers an overview of the structure and genomic architecture of CVB3 (2), mentions the symptoms associated with CVB3 infections (3), addresses the specific CVB3 receptors and their expression in normal tissue as well as in different types of cancers (4), describes the CVB3 strains used as OVs (5), describes the role of CVB3 in tumor microenvironment. An important aspect of the review is the section devoted to improving CVB3 safety by introducing specific microRNA target sites which block the replication of the virus in healthy tissue (7). Finally, the authors mention the use of virus evolution to adapt CVB3 to different types of cancers (8), and the efforts to arm CVB3 with tumor toxic genes (9).

Overall, the manuscript will offer a valuable help to virologists interested in the use of OVs and it paints a meticulous landscape of the research done in this area with CVB3 as a model. Additionally, the authors are visible in the OVs field and their view on this matter is of particular importance. In conclusion, I recommend the article to be accepted.

Comments

I recommend the authors to have a more careful look at the text and correct minor mistake. For instance:

  • lines 4-9 have different types of fonts.
  • line 209: can infect not ‘can infected’
  • line 384: plus-strand not ‘plus-stand’
  • In figure 1 and 3 the white triangle depicts the asymmetric unit and not the protomer.
  • To be correct, the protomer is the assembly of VP0, VP3 and VP1. After the cleavage of VP0 into VP4 and VP2 following the genome encapsidation, the assemble of VP1-4 cannot be properly called protomer. I suggest the authors to correct this in the text in section 2.
  • In Figure 2, in picornaviruses a main role of 2C in that of a helicase.
  • In Figure 3, I suggest the authors to renounce the lower panel in B with a meaning  too obvious to deserve so much typographical space. This would free some space to enlarge the two asymmetric units presented in 2A which are completely unreadable at the moment.

Author Response

Response to Reviewer 1 Comments

Question: recommend the authors to have a more careful look at the text and correct minor mistake. For instance:

  • lines 4-9 have different types of fonts.
  • line 209: can infect not ‘can infected’
  • line 384: plus-strand not ‘plus-stand’

Answer: The manuscript was carefully read, and mistakes have been corrected. According to the list above the changes can be found in the revised manuscript in line 7 and 8; in line 195, and in line 368.

Question: In figure 1 and 3 the white triangle depicts the asymmetric unit and not the protomer. To be correct, the protomer is the assembly of VP0, VP3 and VP1. After the cleavage of VP0 into VP4 and VP2 following the genome encapsidation, the assemble of VP1-4 cannot be properly called protomer. I suggest the authors to correct this in the text in section 2.

Answer: We thank you for this suggestion. In Fig. 1 we replaced “protomer” with “asymmetric unit”. In the text we replaced “protomers” with “asymmetric units” in section 2, line 75, in the Figure legend of Fig. 1 (lines 569, 571 and 574) and in Figure legend of Fig. 3 (lines 590 and 593)

Question: In Figure 2, in picornaviruses a main role of 2C in that of a helicase.

Answer: We are grateful for this important additional information. We inserted the helicase function into Fig. 2 and inserted a sentence in the text on lines 129-130.

Question: In Figure 3, I suggest the authors to renounce the lower panel in B with a meaning  too obvious to deserve so much typographical space. This would free some space to enlarge the two asymmetric units presented in 2A which are completely unreadable at the moment.

Answer: To solve this problem the Fig. 3B was converted into a new Fig. 4, allowing Fig. 3A (new Fig. 3) to be enlarged.

Reviewer 2 Report

Geisler et al. have submitted a very good review of the potential suitability of coxsackievirus B3 for development as an oncolytic virus (OV). Many different viruses are being explored, with encouraging though still limited efficacy of 10-20%. CVB3 displays variable toxicity (depending on strain and other factors) to healthy tissue and in some cases targets tumours efficiently. The review considers existing literature on viral structure and targeting pathways and focuses on engineering improvements by the introduction of miRNA targeting sites and (briefly) the introduction of coding sequences that might enhance viral toxicity in tumour tissue.

The review is very well written and thoroughly referenced, displaying an impressive command of the literature. Curiously, two notable deficiencies are observed. First, almost no consideration is given to the extensive work already underway with Type A coxsackieviruses (especially CVA21 by Viralytics in Australia). This is in advanced trials and already has orphan drug status, has fewer off-target effects or toxicities compared with CVB3, and is otherwise closely related. Surely a comparison of the two virus types (and/or other CVs) would be informative regarding pathways of action and strategies for optimization. A discussion section should be added to the review devoted to this topic.

Second, the review only glancingly considers variations in tumour phenotypes that might make them more or less susceptible to CVB3. Would it be possible, by analyzing tumour biopsies, to identify those patients with tumours most likely to be susceptible to viral oncolysis? Again, a few paragraphs on this strategy, which could allow better targeting of therapy to responsive patients, would broaden and enhance the quality of this review.

Minor editorial notes: line 214 that; line 224 and; line 245 derivate; and lines 321-2 improper reference format.

Overall this is a very good review, well illustrated in its figures. With relatively minor additions it can be excellent. Acceptance for publication is recommended with these changes.

Author Response

Response to Reviewer 2 Comments

Question: Curiously, two notable deficiencies are observed. First, almost no consideration is given to the extensive work already underway with Type A coxsackieviruses (especially CVA21 by Viralytics in Australia). This is in advanced trials and already has orphan drug status, has fewer off-target effects or toxicities compared with CVB3, and is otherwise closely related. Surely a comparison of the two virus types (and/or other CVs) would be informative regarding pathways of action and strategies for optimization. A discussion section should be added to the review devoted to this topic.

Answer: We thank for this suggestion. Our manuscript focuses on CVB3, but we believe that comparison to CVA21 is important, as this virus belongs to the same virus family but is already in clinical trials. We inserted a paragraph (page 10-11, line 481-510) where we compared CVA21 and CVB3.

Question: Second, the review only glancingly considers variations in tumour phenotypes that might make them more or less susceptible to CVB3. Would it be possible, by analyzing tumour biopsies, to identify those patients with tumours most likely to be susceptible to viral oncolysis? Again, a few paragraphs on this strategy, which could allow better targeting of therapy to responsive patients, would broaden and enhance the quality of this review.

Answer: The reviewer raises a very important point, as indeed tumors of the same type may differ from patient to patient and therefore be more or less susceptible to treatment with oncolytic viruses. We believe that pre-selection may be possible, but it may be difficult to do so based solely on the basis of the analysis of the expression of cellular genes from tumor biopsies of the patients, e.g., of the viral receptors (CVB3 and CVA21 use different receptors). Post-entry factors also play a role for replication and oncolytic efficiency of the viruses. Therefore, we believe that further analysis determining the oncolytic activity of the viruses in human tumors material would be necessary to make a final pre-selection. We inserted this assessment at page 11, line 505 to 510

Question: Minor editorial notes: line 214 that; line 224 and; line 245 derivate; and lines 321-2 improper reference format.

Answer: The mistakes were corrected. The changes  can be found in the revised manuscript in line 206, 212, 237 and 309.